

# Identification and validation of hub genes involved in foam cell formation and atherosclerosis development *via* bioinformatics

Da Teng[1,2,*], Hongping Chen[3,*], Wenjuan Jia[1,2], Qingmiao Ren[4], Xiaoning Ding[1], Lihui Zhang[1,2], Lei Gong[1], Hua Wang[1], Lin Zhong[1] and Jun Yang[1,2]

[1] Yantai Yuhuangding Hospital Affiliated to Qingdao University, Yantai, China
[2] Qingdao University, Qingdao, China
[3] Department of Cardiology, The Affiliated Hospital of Xuzhou Medical University, Xuzhou, China
[4] The Precision Medicine Laboratory, The First Hospital of Lanzhou University, Lanzhou, China
[*] These authors contributed equally to this work.

## ABSTRACT

**Background**. Foam cells play crucial roles in all phases of atherosclerosis. However, until now, the specific mechanisms by which these foam cells contribute to atherosclerosis remain unclear. We aimed to identify novel foam cell biomarkers and interventional targets for atherosclerosis, characterizing their potential mechanisms in the progression of atherosclerosis.

**Methods**. Microarray data of atherosclerosis and foam cells were downloaded from the Gene Expression Omnibus (GEO) database. Differentially expression genes (DEGs) were screened using the "LIMMA" package in R software. The Kyoto Encyclopedia of Genes and Genomes (KEGG) pathway enrichment analysis and Gene Ontology (GO) annotation were both carried out. Hub genes were found in Cytoscape after a protein-protein interaction (PPI) enrichment analysis was carried out. Validation of important genes in the GSE41571 dataset, cellular assays, and tissue samples.

**Results**. A total of 407 DEGs in atherosclerosis and 219 DEGs in foam cells were identified, and the DEGs in atherosclerosis were mainly involved in cell proliferation and differentiation. CSF1R and PLAUR were identified as common hub genes and validated in GSE41571. In addition, we also found that the expression of CSF1R and PLAUR gradually increased with the accumulation of lipids and disease progression in cell and tissue experiments.

**Conclusion**. CSF1R and PLAUR are key hub genes of foam cells and may play an important role in the biological process of atherosclerosis. These results advance our understanding of the mechanism behind atherosclerosis and potential therapeutic targets for future development.

Corresponding authors
Lin Zhong, yizun1971@126.com
Jun Yang, yangjyhd@163.com

## INTRODUCTION

According to The 2019 Global Burden of Disease (GBD) Study, global cardiovascular disease (CVD) prevalence increased by 93% from 1990 to 2019, and the number of global

deaths due to CVD rose from 12.1 million in 1990 to 18.6 million in 2019 (*Roth et al., 2020*). Cardiovascular disease is a major cause of the global surge in deaths (*Song et al., 2020*). Atherosclerosis is one of the most common cardiovascular diseases, and its mechanism is complex and diverse, involving but not limited to lipid disorders, endothelial dysfunction, inflammatory cell infiltration, vascular smooth muscle cell differentiation, and other aspects (*Libby et al., 2019*; *Weber & Noels, 2011*). In recent decades, significant progress has been made in preventing, diagnosing, and treating atherosclerosis. However, the molecular mechanisms behind atherosclerosis still need further investigation.

Recently, accumulating evidence has indicated that macrophages play an important role in all stages of atherosclerosis development and progression and are regarded as vital therapeutic targets (*Cho et al., 2013*; *Shaikh et al., 2012*). These macrophages are derived from blood monocytes that, after their recruitment into plaque, differentiate and are activated in response to different environmental signals (*Kim, Ivanov & Williams, 2020*; *Olivares, González Ballester & Noailly, 2016*). Macrophages further evolve into lipid-loaded macrophages by phagocytizing lipids, becoming foam cells. It is remarkable that the formation and accumulation of foam cells are key steps in the development and progression of atherosclerosis (*Hilgendorf, Swirski & Robbins, 2015*; *Moore, Sheedy & Fisher, 2013*). Many drug studies have shown that high-intensity treatment by reducing the level of blood lipids to reduce the number of foam cells engendered many potential therapeutic results (*Allahverdian, Pannu & Francis, 2012*; *Maguire, Pearce & Xiao, 2019*; *Phang et al., 2020*). At this stage, there is still no effective means to detect, monitor and control foam cells. Therefore, it is particularly important to find the precise genetic targets of foam cells in atherosclerosis and intervene.

In recent years, against the backdrop of the big data era, the development of new technologies, such as RNA-seq and microarray expression, have allowed researchers to secondary analyze the data in the database to explore more potential genes related to diseases (*Li et al., 2022*; *Xu & Yang, 2023*). These methods have since become a key one in contemporary biomedical research. One of them, the GEO public database, is frequently utilized to investigate novel pathogenic pathways and disease candidate genes. *Zheng et al. (2023)* analyzed datasets including GSE43292 and found that the key genes—CD52 and IL1RN—were correlated with the infiltration of atherosclerotic immune cells and were also expressed at a high level in foam cells. Through bioinformatic analysis, *Wang et al. (2022)* employed bioinformatic analysis to demonstrate that M1 macrophage infiltration is a significant contributor to plaque instability and that the diagnostic markers CD68, PAM, and IGFBP6 could be used to accurately detect unstable plaques. Genes involved in lipid metabolism, such as PDGTS, DGKE, and others, have been reported by *Liao et al. (2022)* to be significantly correlated with the clinical traits of people with coronary artery disease. Therefore, more data mining using bioinformatics analysis may be able to identify additional probable disease-related mechanisms, offer fresh, workable diagnostic indicators, and contribute to existing science research.

Here, we identified differently expression genes (DEGs) using bioinformatics and functional studies, and exploring the underlying cellular mechanisms may provide

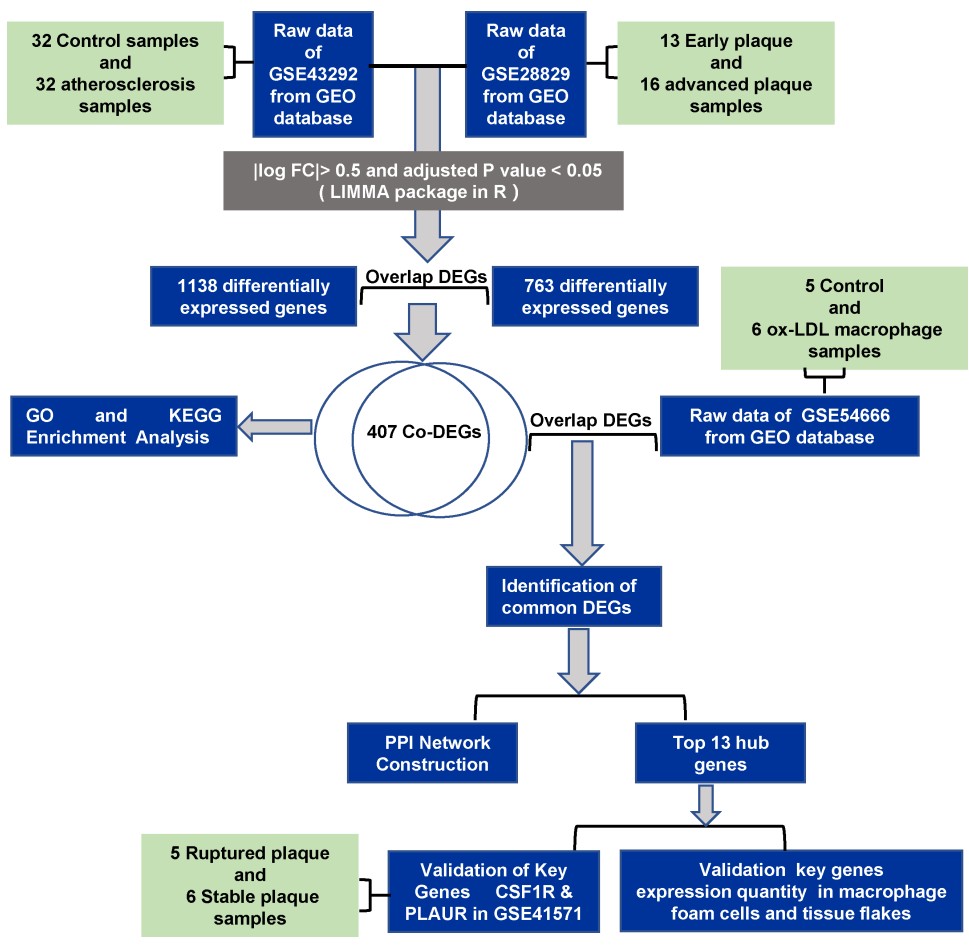

**Figure 1** **The flowchart of the study.**

molecular targets for developing innovative therapies to treat atherosclerosis. The flow diagram of the work is shown in Fig. 1.

## METHODS AND MATERIALS

### Microarray datasets

The Gene Expression Omnibus (GEO) database (http://www.ncbi.nlm.nih.gov/geo) is a gene expression database including data from microarray, gene expression, and gene chip analyses. For this study, we obtained genes associated with the formation and development of atherosclerosis after analysis of two expression profile datasets (GSE28829 and GSE43292) downloaded from GEO. Then, we identified foam cell related DEGs by analyzing the *in vitro* dataset containing ox-LDL treated with macrophages (GSE54666). Finally, we validated our results with GSE41571 and *in vitro* and *in vivo* experiments. In this study, we excluded datasets that contained animal samples and *in vivo* human datasets of serum and plasma.

**Table 1  GEO dataset information.**

| GEO ID | Platform | Samples | Group |
|--------|----------|---------|-------|
| GSE28829 | GPL570 | Atherosclerotic plaque (16 advanced and 13 early) | Discovery set |
| GSE43292 | GPL6244 | Atherosclerotic plaque (32 control and 32 Atherosclerosis) | Discovery set |
| GSE54666 | GPL10588 | Macrophage foam cells (6 control and 6 foam cells) | Discovery set |
| GSE41571 | GPL570 | Atherosclerotic plaque (5 stable and 6 ruptured) | Validation set |

GSE28829 (*Döring et al., 2012*) contains 13 postmortem early atherosclerotic plaques and 16 postmortem advanced atherosclerotic plaques, detected by the Affymetrix Human Genome U133 Plus 2.0 Array. GSE43292 (*Ayari & Bricca, 2013*) is a dataset containing 32 normal carotid artery samples and 32 corresponding atherosclerotic plaque samples, detected by the Affymetrix Human Gene 1.0 ST Array. The GSE54666 (*Reschen et al., 2015*) datasets includes six samples of untreated macrophages and 6 samples of macrophage-derived foam cells stimulated with ox-LDL for 48 h, detected by the Illumina HumanHT-12 V4.0 expression bead chip. GSE41571 (*Lee et al., 2013*) contains data on 11 macrophage-rich regions from five ruptured plaques and six stable plaques, detected by the Affymetrix Human Genome U133 Plus 2.0 Array.

In this study, we selected GSE28829, GSE43292, and GSE54666 as the discovery set. GSE41571 was paired as a validation set for DEG analysis (Table 1).

## Validation of gene expression through DEG analysis

The downloaded gene expression profiles and their matched platform files were loaded into R (version 4.0) software and converted into gene symbol expression profiles. Differential gene expression (differential expression genes, DEG) was performed with the "LIMMA" package (3.50.0) with |log FC| > 0.5 and adjusted $P$ value < 0.05 (*Ritchie et al., 2015*). Co-DEGs were found in the overlap of different datasets as determined by an online web tool (https://www.xiantao.love/).

## Functional enrichment analysis

To elucidate the potential biological mechanisms of genes related to foam cells and atherosclerotic plaques, Gene Ontology (GO) and Kyoto Encyclopedia of Genes and Genomes (KEGG) enrichment analyses were performed using the DAVID website (https://david.ncifcrf.gov/) (*Yu et al., 2012*). At the molecular level, KEGG specifically integrates a large number of practical program database resources from high-throughput experimental technologies. Gene Ontology (GO) is a widely used ontology in bioinformatics. In this study, we analyze two aspects of biology: molecular function (MF), and biological process (BP). The level of statistical significance was set at adjusted $P$ value < 0.05.

## PPI network analysis

The STRING database integrates known and predicted pro-association data for a large number of organisms (https://string-db.org/). We imported the final filtered DEGs into the STRING online database and set the filtering condition as "minimum required interaction score ≥ 0.4" as the threshold value to remove the isolated protein nodes and search for interactions between DEGs encoded proteins. The results are then imported into Cytoscape

software (version: 3.8.0, https://cytoscape.org/) in tsv format to complete the PPI network construction. The Cytoscape MCODE plug-in was used to classify the significant gene modules (clusters), and the parameters were set to the default levels (node degree 2, node score 0.2). Then the Cytohubba plug-in was used to screen out the key genes in the PPI network by the topology network algorithm.

## Validation of the expression of common hub genes in foam cells

The full RPMI-1640 medium was used to cultivate the human monocytic cell line THP-1, which was bought from the American Type Culture Collection (ATCC). This medium also contained 0.05 mM mercaptoethanol and 10% fetal bovine serum. The cells were subcultured at 80–90% confluence after being grown at 37 °C in 5% CO2. The PMA (Merck) concentration of 100 ng/ml was used to induce the differentiation of THP-1 cells into macrophages for 24 h. To create foam cells, the macrophages were treated with 25 or 50 g/ml ox-LDL for 48 h. Oil Red O staining was used to evaluate foam cells.

## Oil red O (ORO) staining

After various interventions, macrophages were washed with PBS three times, fixed with 4% paraformaldehyde for 30 min, washed with PBS twice, and stained with ORO solution for 10 min. The cells were washed with 60% isopropanol for 10 s. Then, the sections were stained with hematoxylin for one minute and washed with deionized water for 10 min. ORO staining was observed under a light microscope.

## Reverse transcription- quantitative PCR (RT-qPCR)

The manufacturer's (Vazyme) recommended procedure for using the TRIzol reagent to isolate total RNA was followed. According to the manufacturer's recommendations, the RNA was reverse-transcribed using a Hifair III First Strand cDNA System (Yeasen Biotechnology). In the PCR amplification, ChamQ Universal SYBR Qpcr Master mix (Applied Biosystems, Vazyme) was used together with 1.0 ul of cDNA and SYBR Green RT-PCR Master Mix. The expression standards of each candidate gene were compared to GAPDH, the internal standard. Using the $2^{-\triangle\triangle Ct}$ method, relative quantitative data were obtained. Each test was carried out three times. The primers are shown in Supplement 1.

## Western blot

We extracted proteins and performed bicinchoninic acid assays according to KeyGEN BioTECH (CAT.NO: KGP902). In brief, 40 ug of proteins were added to an 8% sodium dodecyl sulfate-polyacrylamide gels and then separated by electrophoresis, after which the proteins in the gels were transferred to polyvinylidene fluoride membranes. Blocking the non-specific binding of membranes and incubation with primary antibodies against CSF1R (1:500; Santa Cruz Biotechnology, Dallas TX, USA), PLAUR (1:750; ABclonal, Wuhan, China), and GAPDH (1:10000; Proteintech, China). Incubation with secondary antibodies for 1 h at room temperature after completion. Bands were detected by ECL chemiluminescence (Vazyme, Nanjing Shi, China).

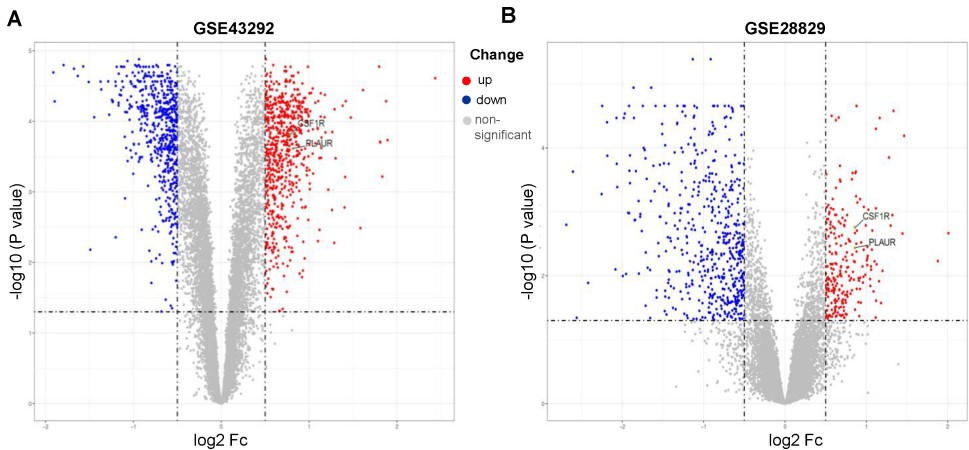

**Figure 2** **Volcano plots of differentially expression genes.** (A) GSE43292, (B) GSE28829. Data points in red represent up-regulated, and blue represent down-regulated genes. The differences are set as an adjusted *P* value < 0.05.

## Immunohistochemical staining

Detection of CSF1R and PLAUR expression in tissue specimens by immunohistochemistry. 5 μm-thick specimens of carotid artery exfoliated tissue were blocked with donkey serum for 1 h at room temperature. Overnight at 4 °C, primary antibodies were administered. Slides were washed in TBS 3 and treated with secondary antibodies before diaminobenzidine development (Thermo Fisher Scientific). CSF1R (1:50; Santa Cruz Biotechnology, USA) and PLAUR (1:50; Santa Cruz Biotechnology, USA) were primary antibodies. Photographs taken with Leica Microsystems (Mannheim, Germany)

## Statistical analysis

Experimental data were analyzed with GraphPad Prism 9.0 (San Diego, CA, USA) Statistically significant differences between groups were calculated by Student's two-tailed *t*-test and $p < 0.05$ was considered to be significant.

## RESULTS

### Identification of DEGs associated with atherosclerosis

GSE43292 contains data on 32 nonatherosclerotic plaque samples (control) and 32 atherosclerotic plaque samples (atherosclerosis). We identified 1,138 DEGs between atherosclerosis and control groups (Supplement 2). Among these DEGs, 663 genes were upregulated and 575 genes were downregulated. The DEGs are shown in the volcano plot and the heatmap whose clustering was performed with Euclidean distance (Figs. 2A and 3A). Simultaneously, we screened 763 DEGs in the GSE28829 dataset, including 236 up- and 527 downregulated DEGs in atherosclerosis, based on gene expression in early plaques (Supplement 3). The DEGs are shown in the volcano plot (Fig. 2B) and heatmap (Fig. 3B). A Venn diagram revealed that the two datasets shared 407 DEGs after comparing the DEGs in the two datasets (Fig. 3C).

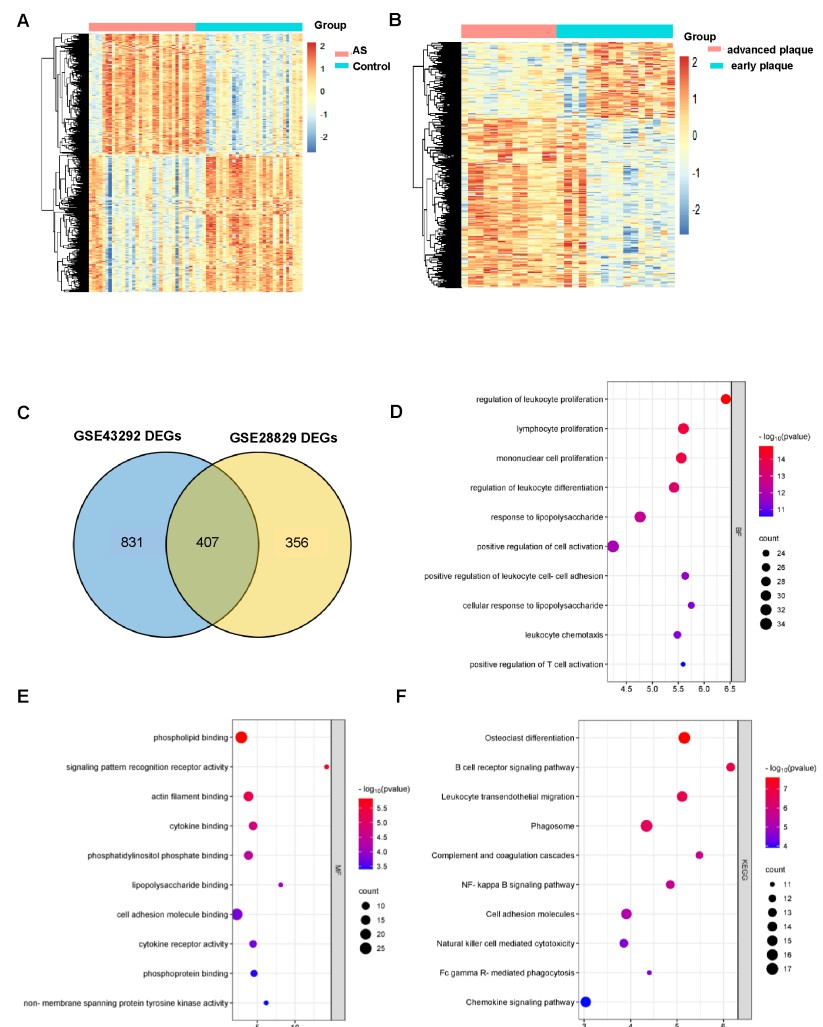

**Figure 3** **AS-DEGs screening and functional enrichment.** Heatmap of differentially expression genes identified in (A) GSE43292, (B) GSE28829 datasets. (C) Venn diagrams showing the co-expression in the GSE43292 and GSE28829 datasets, which were defined as atherosclerosis-related genes. (D) and (E) The bubble diagram of GO-BP and GO-MF enrichment analyses of atherosclerosis-related DEGs. (F) The bubble diagram of the KEGG pathway enrichment analyses of atherosclerosis-related DEGs.

## GO and KEGG analysis of DEGs in atherosclerosis

An improved understanding of atherosclerosis is necessary to advance treatment development. KEGG and GO analyses were performed on all the DEGs to fully explore their biological roles. As shown, all these DEGs were mainly enriched in cell proliferation and differentiation (Fig. 3D). The MF results indicated that the DEGs may be associated with phospholipid binging, signaling pattern recognition receptor activity, actin filament binding, cytokine binding and phosphatidylinositol phosphate binding (Fig. 3E).

All of the differentially expression genes were connected to biological processes through 39 KEGG pathways, according to the findings of the KEGG pathway analysis. The first 10 KEGG pathways are displayed in Fig. 3F.

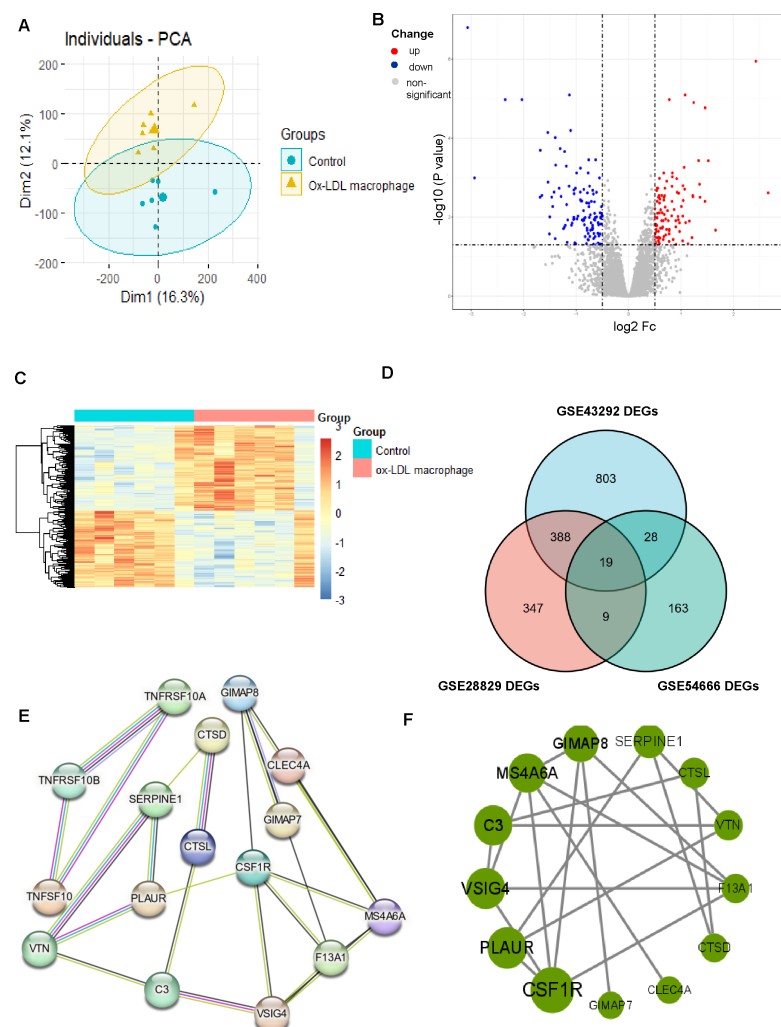

**Figure 4  Identification of DEGs associated with macrophage foam cells and Protein-protein interaction (PPI) network.** (A) PCA score plots of ox-LDL macrophage foam cells group and control group in the dataset GSE54666 (ANOSIM statistic R: 0.3037; p: 0.005). (B, C) Volcano plot (B) and Heatmap (C) of DEGs between ox-LDL macrophage foam cells group and control group. (D) Venn diagrams showing the co-expression in three datasets, which were defined as macrophage foam cells related genes. (E) PPI network of differentially co-expression genes in three datasets. (F) PPI network of the 13 outstanding genes.

## Identification of DEGs associated with macrophage-derived foam cells

The results of the enrichment analysis revealed that cell division, proliferation, and phagocytic activities were the main foci of the DEGs in atherosclerosis. Therefore, we looked into the role the AS macrophage phenotype played. Given the critical role that macrophages play in the formation of atherosclerosis, we obtained the GSE54666 dataset from the GEO database. The GSE54666 gene expression profile depicts the state of foam cells in atheroma plaques in macrophages that have been activated with ox-LDL in vitro. Since the distances between the samples in the control group and the macrophage foam

cell group were both fairly close, the PCA of GSE54666 showed that the 11 samples in the two groups could be separated (Fig. 4A). A total of 219 genes with differential expression, including 104 upregulated and 115 downregulated genes, were found (Supplement 4). The DEGs are shown on both the volcano plot (Fig. 4B) and the heatmap (Fig. 4C). Combining these DEGs with previously identified AS-associated genes led to the identification of overlapped genes as genes involved in foam cell development. As can be seen in Fig. 4D, a total of 19 genes connected to foam cells have been identified.

## Construction of the PPI network and screening of hub genes

The DEGs were organized into a PPI network (Fig. 4E). CSF1R and PLAUR, two of the 13 exceptional genes discovered by PPI analysis and Cytoscape that were classified as hub genes and the subject of subsequent investigations (Fig. 4F), were in the most crucial places.

## Increased expression of hub genes in vulnerable plaques and macrophage-derived foam cells

The validation of the CSF1R and PLAUR expressions then took place utilizing a different dataset. In isolated macrophage-rich areas of stable and ruptured human atherosclerotic plaques, GSE41571 conducted genome-wide expression studies. The hub genes were elevated in the ruptured atherosclerotic plaques, as seen in Fig. 5A.

According to the study, OX-LDL encourages macrophages to take up lipids and create foam cells, which are characteristic of atherosclerotic plaques. A dye called oil red O intensely stains lipids. In order to analyze the accumulation of lipids in macrophage foam cells quantitatively, Oil red O staining was employed (Fig. 5B). The mRNA expression levels of CSF1R and PLAUR were significantly elevated after ox-LDL stimulation in macrophages, according to the RT-qPCR data (Fig. 5C). More crucially, the outcomes showed that the expression level of hub genes increased with the accumulation of lipids. In order to further validate our RT-qPCR findings, we performed additional validation at the protein level. As with our RT-qPCR findings, we found that the expression of CSF1R and PLAUR increased as the number of lipids increased (Figs. 5D to 5G).

## Enhanced expression of hub genes in plaque tissue

We collected surgical specimens from patients undergoing carotid artery dissection and performed immunohistochemical staining of the specimens after paraffin sectioning to detect the expression of hub genes. Similar to the results of our cellular experiments, the expression of CSF1R (Figs. 6A–6B) and PLAUR (Figs. 6C–6D) was increased in atherosclerotic plaques compared to non-plaques.

## DISCUSSION

Atherosclerosis is one of the most important diseases affecting people's health, and its incidence is rising at an alarming rate (Tsao et al., 2022). The mechanism of AS is complex and poorly understood. Among them, the theory of "lipid infiltration" and "inflammatory response" is widely accepted. The molecular mechanisms of foam cells have received attention recently, and several mechanistic investigations have demonstrated that macrophages play a crucial role in the development of AS (Poznyak et al., 2020;

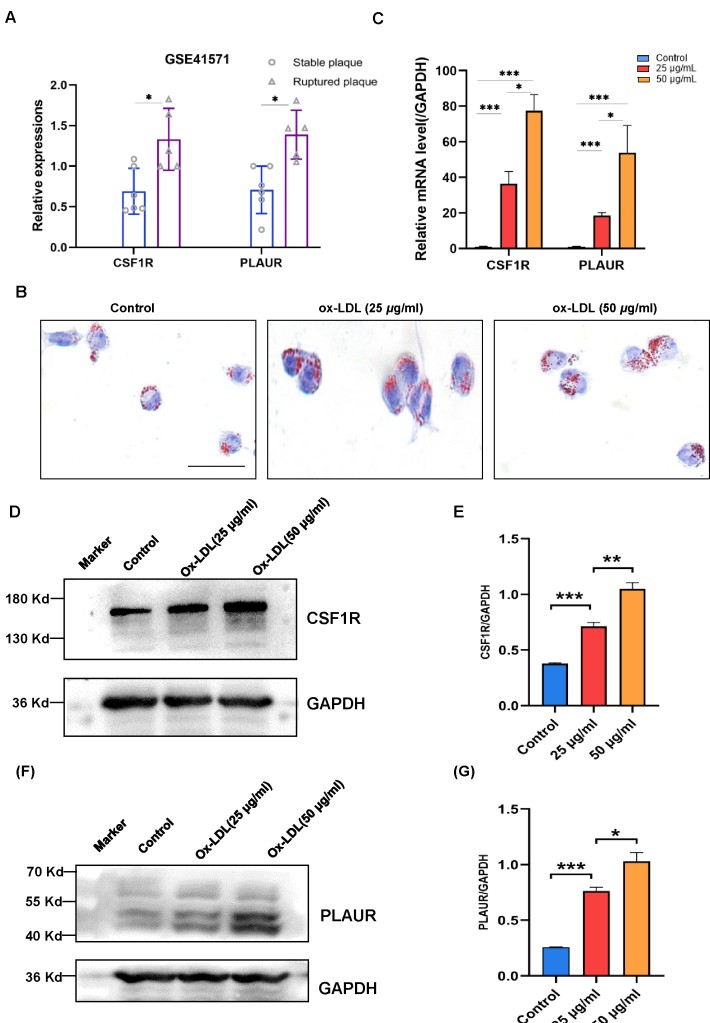

**Figure 5** **The expression levels of CSF1R and PLAUR in the GSE41571 and the ox-LDL-induced macrophage.** (A) The expression levels of hub genes in the GSE41571 (B) Representative Oil Red O staining of macrophage. (C) The mRNA expression levels of common hub genes in ox-LDL-induced macrophage. (D–E) Representative western blots and relative quantitative analysis of CSF1R in macrophages treated with ox-LDL (25 μg/ml, 50 μg/ml) (F–G) Representative western blots and relative quantitative analysis of PLAUR in macrophages treated with ox-LDL(25 μg/ml, 50 μg/ml) (Results are mean ± SEM. *$p < 0.05$, ***$p < 0.01$ compared between the groups by $t$-test. Scale bar: 200 μm).

*Xie et al., 2022*). Our group has also previously found that intervention on foam cells is an effective means of alleviating the atherosclerotic process (*Chen et al., 2023a*; *Chen et al., 2023b*). On the one hand, one of the key initiating events in AS is the infiltration and deposition of cholesterol-rich lipoproteins within the intima of the arterial vessel wall, which results in intimal thickening and narrowing of the artery luminal space. Foam cells are created when infiltrated monocytes differentiate into macrophages, which ingest oxidized LDL and other lipids. This process aids in the development of AS (*Mo et al., 2017*; *Ruiz-León et al., 2019*). On the other hand, atherosclerosis plaques are rich in

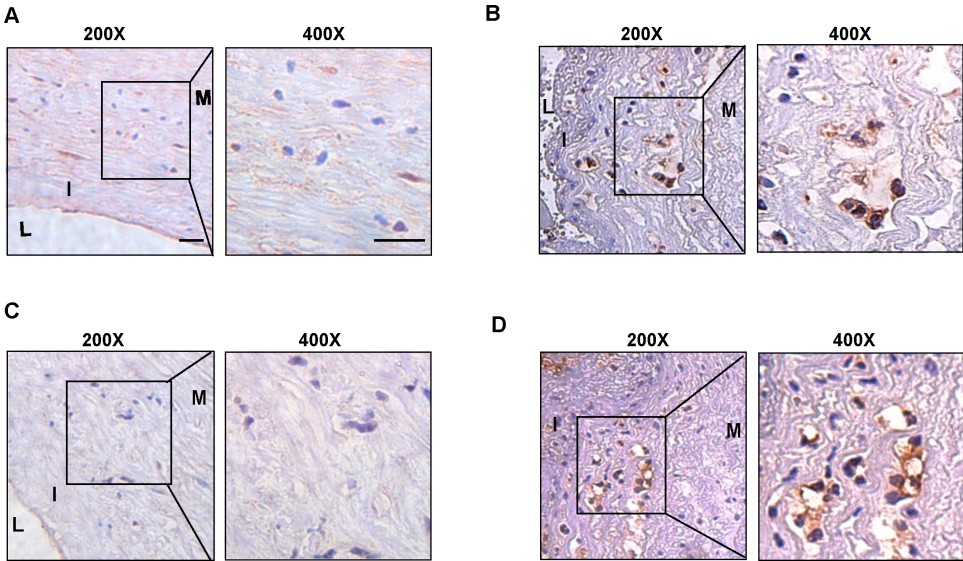

**Figure 6  The expression level of CSF1R and PLAUR in the clinical samples.** (A) The immunohisto-chemistry (200X, 400X) of CSF1R in non-atherosclerotic samples. (B) The immunohistochemistry (200X, 400X) of CSF1R in atherosclerotic samples. (C) The immunohistochemistry (200X, 400X) of PLAUR in non-atherosclerotic samples. (D) The immunohistochemistry (200X, 400X) of PLAUR in atherosclerotic samples. (Scale bar: 100 μm).

foam cells that secrete various pro-inflammatory factors such as IL-1, IL-2, and tumor necrosis factor, which further aggravate the inflammatory response at the lesion and rupture the plaque (*Baker, Hayden & Ghosh, 2011*; *Bobryshev et al., 2016*; *Peng et al., 2020*; *Zhong et al., 2020*). In order to prevent the development of AS, it is crucial to promptly address the lipid buildup in macrophages and the inflammatory response. Based on the above information, we hypothesize that a thorough comprehension of the precise molecular mechanisms of macrophages in AS is essential for the discovery of novel therapeutic targets and the improvement of therapeutic approaches.

The rapid growth of bioinformatics offers a novel approach to the study of atherosclerosis, and mining valuable sequence information from large amounts of data is a fundamental issue in bioinformatics. *Mao et al. (2017)* used bioinformatics to screen top genes related in the development of atherosclerosis, including CXCL12. *Zhou et al. (2022)* used data mining to show a link between SVEP1 and mortality from coronary atherosclerotic heart disease. In addition, *Zhuang et al. (2023)* discovered a unique pathogenic pathway of atherosclerosis *via* the TOP gene-PCSK9. These studies almost always bring fresh optimism for life science research. Despite the fact that numerous researches have looked at the course of atherosclerotic disease from various angles, certain biological mechanisms, such as foam cells, remain unresolved. As a result, we examined the features of foam cells and their mode of operation in AS in this work. More importantly, we attempted to mitigate these consequences through *in vitro* and *in vivo* investigations due to the complex environment of cells in organisms.

In the present study, two microarray datasets (GSE43293 and GSE28829) were analyzed from the Gene Expression Omnibus (GEO) and identified 407 common differentially expression genes (DEGs) between atherosclerotic plaque and control groups. These differentially expression genes were significantly changed compared with normal samples. Due to the multiple etiologies and mechanisms that characterize atherosclerosis, it is certain that numerous pathological alterations are implicated in the progression of the disease (*Napoli et al., 2012*). In our investigation, GO enrichment analysis of biological processes revealed that atherosclerosis-associated DEGs were primarily enriched in cell proliferation and differentiation. Macrophage proliferation, the main immune cell population in atherosclerosis, has emerged as a crucial factor in the development of atherosclerotic plaque (*Moore, Sheedy & Fisher, 2013*). Furthermore, the differentiation of macrophages into foam cells has become a hallmark of atherogenesis (*Sorci-Thomas & Thomas, 2016*). This is just a side point to say that targeting macrophages will undoubtedly be a powerful target for atherosclerosis treatment. Simultaneously, our MF enrichment analysis showed that DEGs are correlated with phospholipid binging, signaling pattern recognition receptor activity, actin filament binding, cytokine binding and phosphatidylinositol phosphate binding. This also illustrates that numerous critical components, such as proteins, cytokines, and transcription factors, play a role in the disease's development in the local microenvironment of atherogenesis (*Krauss, 2010*; *Napoli et al., 2012*; *Smith & Topol, 2006*). When we analyzed the KEGG pathway enrichment of DEGs, we found that they were predominantly enriched in Osteoclast differentiation, B cell receptor signaling pathway, leukocyte transendothelial migration, and phagosome, most of which were activated by the development of atherosclerosis, a significant portion of this is triggered by the development of atherosclerosis (*Hemme et al., 2023*; *Li et al., 2023*; *Wang, Jiang & Cheng, 2022*; *Wang & Chen, 2022*). Furthermore, the development of phagosomes plays a significant part in the production of foam cells (*Huynh, Gershenzon & Grinstein, 2008*), and phagosome intervention may provide a possible support for foam cell reduction.

After screening 407 Co-DEGs with DEGs of macrophage foam cells, a total of 19 related genes were identified. Subsequently, the interaction associations of the proteins encoded by the DEGs were investigated by a protein-protein interaction network (PPI) network, and two hub genes, CSF1R and PLAUR, showed the highest score. This study suggests that they may have had an important effect on the emergence of AS. Finally, we validated our results with GSE41571 and *in vitro* and *in vivo* experiments.

Colony stimulating factor -1 receptor (CSF-1R), an important tyrosine kinase receptor (RTK), regulates the proliferation, differentiation, and survival of mononuclear phagocyte lineage cells, especially macrophages. Compared to normal tissue, early atherosclerosis exhibits a considerably greater level of CSF1 expression (*Shaposhnik, Wang & Lusis, 2010*; *Sinha et al., 2021*). Additionally, the latest researches also point to a correlation between blood levels of CSF1 and the risk of coronary heart disease (*Feldreich et al., 2020*; *Sjaarda et al., 2018*). Now, our study further complements the previous results by confirming that there is also an increase in the expression of CSF1's receptor, CSF1R, in agreement with *Xu, Chen & Yang, (2022)*. More importantly, CSF1R may play a role in lipid

metabolic homeostasis (*Dergunova et al., 2020*). *Irvine et al. (2009)* demonstrated that CSF-1 upregulates atherosclerosis-associated chemokine expression and enhances the cholesterol biosynthesis pathway to promote disease progression when macrophage foaminess is induced *in vitro* and that the selective CSF1R kinase inhibitor, GW2580, markedly reverses this pathologic process. *In vivo* experiments have also yielded comparable results. Several findings validate that CSF1 null mutation, either on an apolipoprotein E (apo E−/−) or a low-density lipoprotein receptor null (LDLR−/−) background, showed a dramatic reduction in the size of atherosclerotic lesions (*Babamusta et al., 2006*). Thus, CSF1R plays a pivotal role in the development of atherosclerosis, but the specific downstream molecular pathways remain unclear due to the complexity of the mechanism and lack of research. However, we have reason to believe that the overexpression of CSF1R is critical to the development of atherosclerosis, and targeting CSF1R as a therapeutic target may reduce the pathologic cascade of pro-atherosclerotic-related factors in patients, and provide some protection against disease progression in patients with atherosclerosis.

PLAUR, also known as urokinase fibrinogen-activated receptor (SuPAR), found mainly in detergent-soluble membranes, is a glycosylphosphatidylinositol-anchored (GPI) triple structural domain (DI, DII, and DIII) receptor protein encoded by the PLAUR gene (*Stephens et al., 1999*). PLAUR interactions with Monocyte Chemotactic Protein 1, Tumor Necrosis Factor, and Interleukins 1 and 6 are strongly linked to increased inflammatory activity in atherosclerotic plaques. Chronic low-grade inflammation, on the other hand, can maintain a state of high cellular metabolism and increased cellular oxygen consumption, resulting in an imbalance in oxygen delivery to inflamed tissues. By using single-cell analysis, Dai et al. discovered that PLAUR expression was considerably enhanced in atherosclerotic macrophages and co-localized with HIF1A, implying that hypoxia may be an essential mechanism by which PLAUR promotes plaque growth (*Dai & Lin, 2023*). Furthermore, PLAUR can influence foam cell development by increasing the production of fibrinolytic enzymes (*Ganné et al., 1999*). As a result, recent research suggests that an increase in PLAUR is associated not only to the number of clinically important atherosclerotic sites but also to the rate of disease development (*Samman Tahhan et al., 2017*). *Hindy et al. (2022)* discovered a strong association between high PLAUR levels and the development of cardiovascular disease and accelerated atherosclerosis through the multidimensionality of epidemiologic, genetic, and experimental evidence, without accounting for diminished renal function and known risk factors. With the popularity of the DrugMatrix database, medications such as desartan and alprostadil may now have some degree of cardiovascular protection using PLAUR as a fulcrum, providing the basis for new indications for new drugs on the market (*Dai & Lin, 2023*). In conclusion, we believe PLAUR is a reliable target for future cardiovascular disease therapy, although more research is needed to demonstrate PLAUR's unique molecular mechanism of action.

Furthermore, our findings suggest that VSIG4, C3, MS4A6A, and GIMAP8 may influence the progression of atherosclerotic plaque. However, the exact pathogenic mechanism remains unknown, and it is possible that multiple mechanisms, such as macrophage polarization, lipid metabolism, and the complement system, all play a role in disease

progression (*Kiss & Binder, 2022*; *Zhang & Zhang, 2022*). We believe that future researchers will undoubtedly contribute to our findings.

In summary, we report two robust predictors of incident coronary disease, we will further validate these findings in future works, and this may provide directions for targeted therapeutic interventions.

## CONCLUSION

Here, we searched for genes associated with pathology using the atherosclerosis dataset from the Gene Expression Omnibus (GEO). The findings of further experiments and bioinformatic analyses revealed that CSF1R and PLAUR are important genes for the development of atherosclerosis. These findings may provide some basis for future drug development for atherosclerotic diseases and provide new viable potential precision targets for disease treatment.

**Abbreviations**

| | |
|---|---|
| **CAD** | Coronary artery disease |
| **GEO** | Gene Expression Omnibus |
| **DEGs** | Differentially expression genesf |
| **GO** | Gene Ontology |
| **KEGG** | Kyoto Encyclopedia of Genes and Genomes |
| **PCA** | Principal component analysis |
| **STRING** | Search Tool for the Retrieval of Interacting Genes |
| **PPI** | Protein-protein interaction |

### Funding

Funding was provided by the National Natural Science Foundation of China (No. 82200503) and the National Natural Science Foundation of Shandong Province (No. ZR2022MH151 and ZR2022QH237). The funders had no role in study design, data collection and analysis, decision to publish, or preparation of the manuscript.

### Grant Disclosures

The following grant information was disclosed by the authors:
National Natural Science Foundation of China: 82200503.
National Natural Science Foundation of Shandong Province: ZR2022MH151, ZR2022QH237.

### Competing Interests

The authors declare there are no competing interests.

## Author Contributions

- Da Teng conceived and designed the experiments, prepared figures and/or tables, and approved the final draft.
- Hongping Chen conceived and designed the experiments, prepared figures and/or tables, and approved the final draft.
- Wenjuan Jia conceived and designed the experiments, prepared figures and/or tables, and approved the final draft.
- Qingmiao Ren analyzed the data, authored or reviewed drafts of the article, responsible for explaining some of the reviewer's questions during revision, and approved the final draft.
- Xiaoning Ding performed the experiments, authored or reviewed drafts of the article, and approved the final draft.
- Lihui Zhang performed the experiments, authored or reviewed drafts of the article, and approved the final draft.
- Lei Gong analyzed the data, authored or reviewed drafts of the article, and approved the final draft.
- Hua Wang analyzed the data, authored or reviewed drafts of the article, and approved the final draft.
- Lin Zhong analyzed the data, authored or reviewed drafts of the article, and approved the final draft.
- Jun Yang conceived and designed the experiments, performed the experiments, analyzed the data, prepared figures and/or tables, authored or reviewed drafts of the article, and approved the final draft.

## Microarray Data Deposition

The following information was supplied regarding the deposition of microarray data:

The data is available at GEO: GSE28829, GSE43292, GSE54666, GSE41571.

## Data Availability

The raw data are available in the Supplemental File.

## Supplemental Information

Supplemental information for this article can be found online at http://dx.doi.org/10.7717/peerj.16122#supplemental-information.

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
