# Peer review of "Identification and validation of hub genes involved in foam cell formation and atherosclerosis development via bioinformatics"

_PeerJ, doi:10.7717/peerj.16122_

## Round 0.1 · original submission · Major Revisions

I believe that major revisions are needed. Please, follow the suggestions.

Reviewer 1 ·

Basic reporting

In this manuscript, Teng et al. utilized three publicly available microarray expression datasets from GEO, identified differentially expressed genes in atherosclerosis and foam cells, and validated two key hub genes. The article is well written. But in general, more information/details should be included in the introduction to explain the background and specific datasets.

• Please conduct a more thorough literature review (e.g., include more explanations and list few specific examples) on how gene expression data is used in atherosclerosis and foam cells. The authors introduced atherosclerosis and importance of foam cells. They also briefly mentioned about microarray expression data. However, they fail to mention about the current practice of genomics/expression data in the context of atherosclerosis and foam cells. For example, “Single-cell RNA sequencing in atherosclerosis: Mechanism and precision medicine” https://www.frontiersin.org/articles/10.3389/fphar.2022.977490/full provides a comprehensive review from single-cell perspective.

Experimental design

• Please explain the rationale of choosing these four datasets. Are they selected based on certain criteria? Is there any selection bias? Please comment on whether the results are generalizable to other atherosclerosis studies.
• Please provide more details and summarize the background and findings for the four GEO datasets. For example, did the previous publication conduct similar analysis? If so, was there any overlap between the DEGs reported in this article and the original publication, or any difference?
• Please share the code (e.g., github) and specify the parameters/version of the software tools to ensure reproducibility.
• In section 2.3, the authors focused on MF and BP aspects in the GO enrichment pathway analysis using threshold of p<0.05. Considering the multiple testing issue, please use adjusted p-value<threshold in the results instead. Will it affect the results? Also, what’s the result of cellular component (CC)?
• In section 2.4, an online tool was used to perform PPI analysis. What’s the tool? Please include a brief description of the process rather than listing the software name. Also, please specify the parameters and describe how to determine the hub genes quantitively.
• Please provide rationale for intersecting the DEGs inferred from GEO datasets. It seems that the contexts of comparison are different - the first dataset compared control vs atherosclerosis; the second dataset compared early vs late atherosclerosis.
• In Fig3, please annotate and highlight the important genes to be discussed in the volcano plots. Please discuss specific genes in detail about their biological function in the context of atherosclerosis and foam cells.
• Similarly, in addition to listing the significant pathways, please describe the GO and KEGG pathway in the context of atherosclerosis and foam cells. Please cite related references.
• In Fig4, please quantify the separation of two groups in the PCA plot. Random shuffling the sample label multiple times can be used to construct null.
• Please include more details about PPI network: summarize the key steps in words.

Validity of the findings

no comment

Reviewer 2 ·

Basic reporting

Clear and well executed. The authors in the manuscript aim to identify novel foam cell biomarkers and interventional targets for atherosclerosis. The question is of importance and overall the plan is solid. References are fine except for some webtools used. The link is mentioned but I am not sure if they are cited properly. Kindly look into that.
1. The use of some words could be better for example in line 38 in my opinion it is better to use mechanism or manifestation of the disease instead of pathogenesis that refers primarily affected by pathogen.

2. Kindly mention the version used for R/Bioconductor packages used also cite them for example limma. Take care of other tools also in similar way.

3. In figure 1 (flowchart) heading formation needs to be looked at again for example authors can say- 1138 differentially expressed genes instead of saying 1138 different expression genes

4. In fig 1 incorporate where you used Limma . It will be clearer to readers.

5. Figure 2 ID of one dataset is wrongly mentioned

Experimental design

Plan and execution seem solid.

Validity of the findings

The authors talked minimal about top DE genes in the manuscript. Top genes related to the atherosclerosis based on other literature like SVEP1, CXCL12 do not show up at all. I am a bit reluctant here because authors mentioned top 2 hub genes but not about other genes. The authors need to justify this.

Other points-

1. The authors need to state clearly that up or down-regulated genes were in non-control samples. mention that in figure 2 also.

2. I wonder why authors call rest of the genes in fig 2 (gray dots) stable? I understand that these genes are not differentially expressed but I am not sure if calling them stable justifies that. There are many genes that are above the threshold of LFC but they did not qualify statistical significance test criterion. I would prefer to call them non-significant genes in the test rather than overstating by calling them stable.

3. In figure 2 the dotted lines (horizontal and vertical ) are showing the threshold then why are there so many non-significant genes in the significant area of plot?

4. Figure 3 also has a typo in data ID I believe. Kindly recheck throughout the manuscript.
5. Figure 3 A does not clearly show any genes on x-axis. Improve the plots please and mention the genes with their common names rather than IDs.

6. Point 5 also applies to other heatmaps.

7. Revisit the figures (some of plots seem to be stretched from original or weirdly popping up).

---

## Round 0.2 · accepted · Accept

The authors addressed properly all the reviewers' comments.
I approve the current revised version.
The manuscript is now ready for publication.

Reviewer 1 ·

Basic reporting

no comment

Experimental design

no comment

Validity of the findings

no comment